

# Life history and past demography maintain genetic structure, outcrossing rate, contemporary pollen gene flow of an understory herb in a highly fragmented rainforest

Pilar Suárez-Montes, Mariana Chávez-Pesqueira and Juan Núñez-Farfán

Laboratory of Ecological Genetics and Evolution, Department of Evolutionary Ecology, Instituto de Ecología, Universidad Nacional Autónoma de México (UNAM), Mexico

Corresponding author
Juan Núñez-Farfán, farfan@unam.mx

## ABSTRACT

**Introduction**. Theory predicts that habitat fragmentation, by reducing population size and increasing isolation among remnant populations, can alter their genetic diversity and structure. A cascade of effects is expected: genetic drift and inbreeding after a population bottleneck, changes in biotic interactions that may affect, as in the case of plants, pollen dynamics, mating system, reproductive success. The detection of the effects of contemporary habitat fragmentation on the genetic structure of populations are conditioned by the magnitude of change, given the few number of generations since the onset of fragmentation, especially for long-lived organisms. However, the present-day genetic structure of populations may bear the signature of past demography events. Here, we examine the effects of rainforest fragmentation on the genetic diversity, population structure, mating system (outcrossing rate), indirect gene flow and contemporary pollen dynamics in the understory herb *Aphelandra aurantiaca*. Also, we assessed its present-day genetic structure under different past demographic scenarios.

**Methods**. Twelve populations of *A. aurantiaca* were sampled in large (4), medium (3), and small (5) forest fragments in the lowland tropical rainforest at Los Tuxtlas region. Variation at 11 microsatellite loci was assessed in 28–30 reproductive plants per population. In two medium- and two large-size fragments we estimated the density of reproductive plants, and the mating system by analyzing the progeny of different mother plants per population.

**Results**. Despite prevailing habitat fragmentation, populations of *A. aurantiaca* possess high genetic variation ($H_e = 0.61$), weak genetic structure ($R_{st} = 0.037$), and slight inbreeding in small fragments. Effective population sizes ($N_e$) were large, but slightly lower in small fragments. Migrants derive mostly from large and medium size fragments. Gene dispersal is highly restricted but long distance gene dispersal events were detected. *Aphelandra aurantiaca* shows a mixed mating system ($t_m = 0.81$) and the outcrossing rate have not been affected by habitat fragmentation. A strong pollen pool structure was detected due to few effective pollen donors ($N_{ep}$) and low distance pollen movement, pointing that most plants received pollen from close neighbors. Past demographic fluctuations may have affected the present population genetic structure
as Bayesian coalescent analysis revealed the signature of past population expansion, possibly during warmer conditions after the last glacial maximum.

**Discussion**. Habitat fragmentation has not increased genetic differentiation or reduced genetic diversity of *A. aurantiaca* despite dozens of generations since the onset of fragmentation in the region of Los Tuxtlas. Instead, past population expansion is compatible with the lack of observed genetic structure. The predicted negative effects of rainforest fragmentation on genetic diversity and population structure of *A. aurantiaca* seem to have been buffered owing to its large effective populations and long-distance dispersal events. In particular, its mixed-mating system, mostly of outcrossing, suggests high efficiency of pollinators promoting connectivity and reducing inbreeding. However, some results point that the effects of fragmentation are underway, as two small fragments showed higher membership probabilities to their population of origin, suggesting genetic isolation. Our findings underscore the importance of fragment size to maintain genetic connectivity across the landscape.

## INTRODUCTION

Tropical rainforests sustain much of global biodiversity, including most endemic plant species of the world (*Myers, 1988*). Unfortunately, tropical rainforests have been reduced to half of their original area (*FAO, 2014*) and face intense pressures from agriculture and livestock expansion (*Seymour et al., 2014*). Forest fragmentation is, thus, one of the main threats to rainforest biodiversity due to its effects on physical environmental conditions, ecological interactions, and genetic processes (*Young, Boyle & Brown, 1996*; *Haddad, 2015*).

Theorized genetic consequences of habitat fragmentation have focused on effects brought about by reductions in population size and increasing spatial isolation between remnant populations (*Young, Boyle & Brown, 1996*; *Aguilar et al., 2008*). These changes may reduce genetic variability and increase population genetic structure. After sudden reductions of effective population size or recent bottlenecks, genetic drift and inbreeding will cause further loss of alleles—especially rare alleles—thus increasing homozygosity. On the other hand, reduced connectivity among populations, gene flow cannot prevent the loss of alleles leading to genetic structuring. Disruption of gene flow of plant populations inhabiting fragments may modify mating patterns, reducing outcrossing rates and reproductive success, and consequently increase inbreeding. In the long term, these effects may affect fitness of populations, their adaptability to novel environmental conditions, and increasing the risk of local extinction (*Young, Boyle & Brown, 1996*; *Aguilar et al., 2008*; *Breed et al., 2013*; *Finger et al., 2014*).

The impact of rainforest fragmentation on the genetic structure of plant populations is highly variable, depending on life history, life-span, and mating system (*Cuartas-Hernández & Núñez-Farfán, 2006*; *Figueroa-Esquivel et al., 2010*; *Suárez-Montes, Fornoni & Núñez-Farfán, 2011*; *Chávez-Pesqueira et al., 2014*). At the landscape scale, factors like
the spatial configuration of fragments can also explain the genetic change of populations (*Leimu et al., 2006*; *Vranckx et al., 2011*; *Aparicio et al., 2012*; *Chávez-Pesqueira et al., 2014*). Moreover, it is important to consider past demographic processes can impact the patterns of present-day of genetic diversity (*Hsieh et al., 2013*). Therefore, a comprehensive knowledge of life history, population genetic structure, effective population size, and demographic history changes is fundamental to develop conservation strategies aimed to maintain genetic variability and evolutionary potential of plant species across fragmented rain forests.

Herbs represent *ca.* 45% of vascular plant diversity and are the richest plant communities in lowland tropical rainforests (*Gentry Alwyn & Dodson, 1987*; *Parkes, Newell & Cheal, 2003*). Although tropical rainforest herbs may play an important role to maintain forest structure, functioning, and dynamics (*Richards, 1996*), the genetic effects of fragmentation on this life form have not been extensively studied. Tropical plants species represent only 20% of the total species analyzed in fragmentation studies; of these only 4% are herbs, whereas 88% are canopy trees (*Aguilar et al., 2008*; *Vranckx et al., 2011*). Specifically, understory herbs are ideal systems to detect genetic effects of habitat fragmentation on a shorter time-scale, owing to their dependence on canopy cover, and relative short life span in relation to long-lived canopy trees (*Lowe et al., 2005*). Moreover, their natural distribution is exposed to altered ecological and environmental conditions by forest fragmentation, which may modify outcrossing rates, contemporary pollen dynamics and mating patterns.

Very few detailed studies on herbaceous plants have measured contemporary pollen dispersal within and among fragmented populations (*Gonzales et al., 2006*; *Cuartas-Hernández, Núñez-Farfán & Smouse, 2010*; *Côrtes et al., 2013*). Results revealed restricted pollen movement of herbaceous plants with an unclear pattern of the effects of fragmentation. In some cases forest fragmentation has limited impact on pollen dynamics (*Cuartas-Hernández, Núñez-Farfán & Smouse, 2010*), while in others it increases pollen movement and decreases pollen structure, possible due to edge effects (*Gonzales et al., 2006*). Furthermore, since pollen dispersal could be associated to the density of reproductive plants, forest fragmentation can enhance/reduce gene dispersal depending on plant abundance (*Cuartas-Hernández, Núñez-Farfán & Smouse, 2010*; *Breed et al., 2012*; *Côrtes et al., 2013*).

Here, we assess the genetic structure, out-crossing rate, and contemporary pollen dynamics of populations of *Aphelandra aurantiaca* (Acanthaceae) in a highly fragmented tropical rainforest in southern Mexico. Also, using contemporary genetic data, we infer the demographic history to understand the current distribution of genetic diversity. *Aphelandra aurantiaca* is an important species of tropical rainforest understory, whose population dynamics is affected by the presence of sunflecks or forest light-gaps (*Calvo-Irabién, 1989*; *Calvo-Irabién, 1997*; *Calvo-Irabién & Islas-Luna, 1999*). Although species of Acanthaceae are among the most important flowering plants in the forest's understory, studies assessing their genetic diversity are still lacking. To our knowledge, this is the first study that assesses the effects of habitat fragmentation on the genetic structure and contemporary gene flow in a tropical herbaceous plant of the genus *Aphelandra*.

We assessed the potential effects of habitat fragmentation on the genetic structure of *A. aurantiaca* of populations inhabiting fragments of different area. Specifically, we tested the hypothesis that, unlike large or medium sized fragments, small ones will show (1) reduced genetic variation and effective population sizes, (2) higher population differentiation as a consequence of genetic isolation, (3) higher inbreeding, (4) lower out-crossing rate, and (5) higher differentiation among pollen pools, (6) reduced number of pollen donor parents, and (7) decreased effective pollination neighborhood due to a decrease in plant abundance. Complementary, we assessed the present day genetic structure of *A. aurantiaca* under different past demographic scenarios.

## MATERIALS & METHODS

### Study system

*Aphelandra aurantiaca* (Scheidw.) Lindl. is an understory herb of neotropical rainforests from southern Mexico to Bolivia. It is a self-compatible species that bears inflorescences with yellow floral buds that turn red when flowers open and produce nectar. In the rainforest of Los Tuxtlas in Mexico, this species is pollinated by the hummingbird *Phaethornis longirostris* (I Ramírez-Lucho, P Suárez-Montes & J Núñez-Farfán, pers. obs., 2013) although it is also visited by butterflies and bumble-bees (*Calvo-Irabién, 1989*; *Islas Luna, 1995*). Seed dispersal is ballistic, ranging from 1 to 8.5 m (modal value of 1.5 m) from the maternal plant. This herb also exhibit vegetative reproduction by stolons. Its life span ranges from 13 to 18 years (*Calvo-Irabién, 1989*). Reduction in plant abundance of *A. aurantiaca* is related to the forest regeneration cycle where light is the most variable abiotic factor; the species inhabits both shaded forest understory and forest light-gaps (*Calvo-Irabién, 1989*; *Calvo-Irabién, 1997*).

### Study site and data collection

The study was carried out at Los Tuxtlas Biosphere Reserve in southern Mexico, which constitutes the northernmost distributional limit of tropical rainforest in the Americas (*Dirzo & Miranda, 1991*). The region has lost more than 90% of its original forest cover in the past fifty years. Nowadays, the current landscape is composed of areas used for human settlement (1.27%), roads (0.78%), water bodies (1.92%), cattle ranching and crops (42.82%), riparian strips (4.29%), live fences (3.28%), isolated trees (1.03%), secondary vegetation of rainforest (0.53%), and fragments of rainforest (23.24%). Rainforest fragments are relatively small (<100 ha), surrounded by grassland and located in lowlands or restricted to the top of the mountains, in glens or areas of difficult access. At higher elevations (>600 m a.s.l.) cloud forest (4.61%) and secondary cloud forests (0.25%) is the predominant vegetation (*Dirzo & Garcia, 1992*; *CONANP, 2011*; see *Salazar Arteaga, 2015*) (Fig. 1).

Twelve populations of *A. aurantiaca* were sampled throughout Los Tuxtlas rainforest (Table 1). Preserved areas covered by rainforest are mostly surrounded by a matrix of pasture lands used for cattle ranching. Because the study area is highly fragmented, the choice of sampling sites was based on size area, accessibility, and the possibility of getting large sample sizes for genetic analyses (∼30 individuals). Forest fragment size best explains

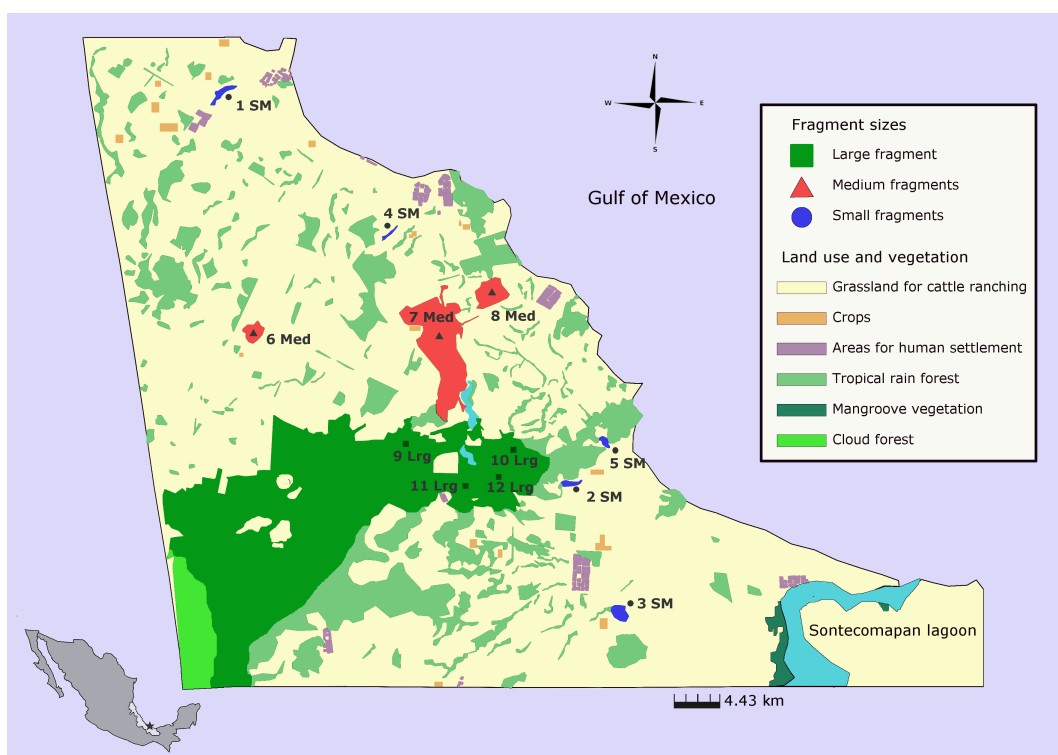

**Figure 1** Populations of *Aphelandra aurantiaca* sampled for genetic analyses at Los Tuxtlas rainforest. Colors represent fragment size classes: blue (small), red (medium), green (large). Names of populations as in Table 1.

the differences in composition and plant structure (*Arroyo-Rodríguez & Mandujano, 2006*) therefore, we classified fragments as small (≤10 ha), medium (20–120 ha) and large and undisturbed fragments (>640 ha) (Fig. 1, Table S1). Unlike medium and large forest fragments, forest structure of small fragments is characterized by the absence of large primary trees in the canopy and lower abundance of palms and some herb species in the understory, but high abundance of shade intolerant secondary species (see *Arroyo-Rodríguez & Mandujano, 2006*). The large, undisturbed and continuous forest is in the core of Los Tuxtlas Biosphere Reserve; however it is partially separated by a deforested area produced by an illegal invasion (see Fig. 1). We established four sites within the large fragment as different populations based on their distance from deforested areas, and geographic distance between sampled sites (from 4. 39 to 14 km).

In each fragment size, we collected young leaf tissue from 29 to 38 adult plants. Individuals were selected as reproductive if they had an inflorescence/infructescence, or scars of these on the stem. We collected leaf tissue from individuals separated at least three meters apart. In total, we sampled individuals of *A. aurantiaca* in five small ($n = 153$), three medium ($n = 97$) and four large fragments ($n = 138$) (Table 1).

## Microsatellites analysis and PCR amplification

We amplified 11 polymorphic microsatellites specifically developed for *A. aurantiaca* to determine its genetic structure (*Suárez-Montes, Tapia-López & Núñez-Farfán, 2015*)

**Table 1** Genetic diversity of twelve populations of *Aphelandra aurantiaca* at the Los Tuxtlas tropical rainforest.

| Size | Fragments | ha | A (s.d.) | P(1) (s.d.) | Ho(s.d.) | He(s.d.) | Fis (C.I.) | Ne (C.I.) |
|---|---|---|---|---|---|---|---|---|
| | 1 SM | 8 | 5.72 (2.10) | 0.25(0.35) | 0.57 (0.12) | 0.66 (0.12) | 0.134 (0.034, 0.201) | 68.2 |
| | 2 SM | 5 | 5.72 (1.79) | 0.25(0.32) | 0.63 (0.21) | 0.63 (0.16) | −0.012 (−0.110, 0.047) | 131.6 |
| Small | 3 SM | 8 | 4.45 (1.96) | 0.20(0.35) | 0.38 (0.24) | 0.54 (0.21) | 0.290 (0.154, 0.361) | 19.2 |
| | 4 SM | 4 | 5.54 (2.54) | 0.18(0.20) | 0.59 (0.19) | 0.61 (0.17) | 0.038 (−0.051, 0.092) | 70.1 |
| | 5 SM | 5 | 5.09 (1.57) | 0.14(0.20) | 0.64 (0.21) | 0.63 (0.14) | −0.012 (−0.105, 0.045) | 30.6 |
| Total small | | | 9.0 (3.39) | 0.74(0.48) | 0.57 (0.16) | 0.65 (0.11) | 0.114 (0.071, 0.150) | 119.2 (86.4–176.1) |
| | 6 Med | 17 | 6.63 (2.94) | 0.27(0.25) | 0.50 (0.14) | 0.64 (0.13) | 0.217 (0.094, 0.312) | 35.7 |
| Medium | 7 Med | 120 | 5.45(3.44) | 0.23(0.33) | 0.52 (0.24) | 0.56 (0.17) | 0.076 (−0.050, 0.176) | 69.9 |
| | 8 Med | 35 | 5.81 (3.86) | 0.21(0.47) | 0.66 (0.19) | 0.63 (0.15) | −0.045 (−0.136, 0.014) | 39.4 |
| Total medium | | | 8.72 (5.98) | 0.65(0.65) | 0.56 (0.16) | 0.63 (0.15) | 0.109 (0.045, 0.16) | 146.7 (89.1–326.3) |
| | 9 Lrg | 640 | 6.27 (2.86) | 0.22(0.25) | 0.611 (0.18) | 0.64 (0.16) | 0.047 (−0.032, 0.101) | infinite |
| Large | 10 Lrg | 640 | 6.27 (2.32) | 0.26(0.28) | 0.55 (0.18) | 0.59 (0.12) | 0.068 (−0.031, 0.138) | 510.1 |
| | 11 Lrg | 640 | 4.81 (1.47) | 0.14(0.25) | 0.64 (0.19) | 0.60 (0.15) | −0.066 (−0.176, 0.014) | 24.7 |
| | 12 Lrg | 640 | 5.09 (2.16) | 0.16(0.15) | 0.53 (0.22) | 0.56 (0.19) | 0.060 (−0.060, 0.144) | 31.4 |
| Total large | | | 8.54 (4.32) | 0.60(0.43) | 0.59 (0.16) | 0.622 (0.14) | 0.051 (0.005, 0.089) | 205.7 (123–480.6) |

**Notes.**

*ha*, hectares by fragment; *A*, number of alleles per locus; *P(1)*, private allelic richness; *Ho*, observed heterozygosity; *He*, expected heterozygosity; *Fis*, inbreeding; *Ne*, effective population size; *(s.d.)*, standard deviation; *C.I*, 95% confidence interval. Total values for each category of fragment size are provided.

(Table S2). DNA extraction, amplification, and laboratory setup are detailed in *Suárez-Montes, Tapia-López & Núñez-Farfán (2015)*. We genotyped all sampled individuals and scored alleles using the software GeneMarker V.2.4.0 (SoftGenetics, State College, PA, USA).

## Genetic diversity analyses

We used MICRO-CHECKER software (*Van Oosterhout et al., 2004*) to detect null alleles. Deviations from Hardy–Weinberg equilibrium (HWE) and linkage disequilibrium were tested using GenePop 4.2 (*Rousset, 2008*) and FSTAT v.2.9.3.2 (*Goudet, 1995*), respectively. We estimated descriptive statistics of genetic diversity including allelic richness (*A*) and expected and observed heterozygosity (*He* and *Ho*) using Arlequin v.3.5.1.3 (*Excoffier, Laval & Schneider, 2005*). The private allelic richness (*P(1)*) was calculated by rarefaction with H$_{P-RARE}$ (*Kalinowski, 2005*). Mean inbreeding coefficient (*F$_{IS}$*) was estimated using GENETIX v.4.05 (*Belkhir et al., 1996–2004*), based on 10,000 permutations. The hypothesis of isolation-by-distance was tested by a Mantel test based on 999 replicates using the ade4 package (*Dray & Dufour, 2007*) in R 3.1.3 (*R Development Core Team, 2015*).

## Population size and demographic history analyses

We estimated effective population sizes (*N$_e$*) using a linkage disequilibrium method in NeEstimator V2 (*Do et al., 2014*). We employed two approaches to detect changes in population size. The first approach is based on the detection of heterozygosity excess or deficiency in a very recent period of time ($2N_e$–$4N_e$ generations) using the program Bottleneck 1.2.02 (*Piry, Luikart & Cornuet, 1999*). Populations recently bottlenecked would lose rare alleles faster than heterozygosity, resulting in an apparent heterozygosity excess in

comparison with a population at equilibrium (*Heq*) (*Cornuet & Luikart, 1997*; *Piry, Luikart & Cornuet, 1999*).

The second approach employs coalescent simulations in an approximate Bayesian computation (ABC) framework to infer past demographic history, as implemented in the software DIYABC 2.0 (*Cornuet et al., 2008*). ABC chooses a demographic scenario that best fits the observed data by running simulations constrained by the specifications of the model (e.g., a demographic bottleneck). It approaches the posterior probability distributions of parameters by selecting the simulated datasets with the smallest Euclidian distances to the observed data, as measured by summary statistics (*Cornuet et al., 2008*; *Cornuet et al., 2014*). We compared four demographic scenarios: two with constant $N_e$, one with a decline, and one with an expansion. We ran one million simulations for each scenario. The parameter settings and priors are shown in Table S3. The change time in $N_e$ was set at 10–10,000 before present, assuming generation time of one year for the species (*Calvo-Irabién, 1989*). We used a generalized stepwise mutation model with a mutation rate of $10^{-4}$–$10^{-3}$. The summary statistics were: mean number of alleles, mean heterozygosity, and mean allelic size variance. We assessed the fit of the models to the data by principal components analysis (PCA), implemented in DIYABC. We estimated the posterior probabilities using a logistic regression approach on the first 1% simulations closest to the observed dataset. To check the confidence of model choice we estimated type I and type II error rates by simulating 500 pseudo-observed data sets (*Cornuet et al., 2008*; *Cornuet et al., 2014*).

## Genetic structure and clustering analyses

We estimated *Rst* and a hierarchical analysis of molecular variance (AMOVA) using Arlequin v.3.5.1.3 program (*Excoffier, Laval & Schneider, 2005*). Population structure was explored with both model based (Structure) (*Pritchard, 2010*) and distance based approaches (DAPC, Pop Graph and NetStruct) (*Jombart, Devillard & Balloux, 2010*; *Greenbaum, Templeton & Bar-David, 2016*).

The Bayesian clustering algorithm implemented in Structure V2.3 (*Pritchard, 2010*) estimates the probability of genotypes being distributed into *K* number of clusters. Simulations were run using correlated allele frequencies, under admixture ancestry models, conducting a burn-in of $10^6$, MCMC iterations of $10^6$, and *K* varying from 1 to 12. The total number of clusters (*K*) was inferred with the Evanno Δ*K* method (*Evanno, Regnaut & Goudet, 2005*) using STRUCTURE HARVESTER (*Earl & vonholdt, 2012*) and CLUMPAK pipeline (Cluster Markov Packager across *K*) (*Kopelman et al., 2015*) to visualize bar plots.

NetStruct 1.2 program (*Greenbaum, Templeton & Bar-David, 2016*) infers genetic structure using network theory. The equivalent of a genetic population structure is the community partition of a network constructed with individuals as nodes and edges (paired connections of nodes), defined by using a similarity measure. Clustering is done by locating groups of nodes (community) that are strongly connected within the group but weakly connected to nodes outside the group. The strength association (SA) measures how strongly the individuals are related to the community at which they were assigned to. The strength association distribution (SAD) analysis examines the distribution of SA values

of different communities and provides information about relative gene flow. A narrow SAD indicates low gene flow, while left-skewed SAD suggests constant moderate gene flow; recent migrants will display low SA values, increasing the variance and left–skewness of the distribution. Finally, NetStruct evaluates the statistical significance of community partitions using permutation tests (*Greenbaum, Templeton & Bar-David, 2016*). We used the Fast Greedy algorithm with a medium threshold of 0.24 and 999 permutations for modularity significance test. We characterized the SAD of communities by the Coefficient of Variation (CV), as a measure of dispersion.

## Membership probability analysis

To evaluate the membership probability of individuals to their population of origin we used the discriminant analysis of principal components (DAPC), a multivariate analysis implemented in the *adegenet* package (*Jombart, 2008*; *Jombart, Devillard & Balloux, 2010*) in R 3.1.3 (*R Development Core Team, 2015*). Based on the retained discriminant functions, the analysis derives membership probabilities for each individual of original source populations (*Jombart & Collins, 2015*). We followed *adegenet* directions for alpha scores optimization; we performed the analysis with 50 PCs retained. We also evaluated admixed individuals with no more than 0.5 probability of membership to any population.

## Spatial structure and genetic barriers analyses

To visualize the spatial genetic structure and the connectivity across the landscape we used Population Graphs implemented in *gstudio* (*Dyer & Nason, 2004*) and *popgraph* packages (*Dyer, 2009*; *Dyer, 2014*) in R 2.15.3 (*R Development Core Team, 2013*). Population Graphs is a graph-theoretical approach where the total genetic variation is decomposed into a geometric interpretation of components within and among the strata, and then modified using conditional covariance to the minimal topological configuration. The genetic structure within population variance is represented as nodes that are connected by edges whose magnitude is proportional to their interpopulation variance (*Dyer, 2015*). Nodes and edges were then mapped on their spatial coordinates. We tested isolation across graph distance (IBGD) through physical and graph distances using *Graph* software of GeneticStudio software (*Dyer, 2009*; *Dyer, 2014*). To identify long distance dispersal events or restricted gene flow we compared pairwise physical distances with their corresponding pairwise edge lengths using *Graph* software (*Dyer, 2009*). Edges whose length is significantly longer than expected indicate long distance dispersal events while edges with length shorter than expected indicate limited dispersal across the landscape (*Dyer, 2015*).

## Recent migration analyses

Recent bidirectional migration rates (in the last 2–5 generations) were estimated for paired populations using the program BayesAss v 3.0 (*Wilson & Rannala, 2003*). Because we expected reduced connectivity for smaller and more isolated fragments, we also estimated migration rates between different fragment sizes. BayesAss does not depend on Hardy–Weinberg equilibrium and estimates $m$ as the fraction of individuals in population $i$ that are migrants derived from population $j$ (per generation) (*Wilson & Rannala, 2003*). To check for consistency, we performed 10 runs with a different random seed number, and

calculated their respective Bayesian deviance (*Meirmans, 2014*). The run with the lowest deviance value was used to select the best-fitting model. Each run consisted of three million iterations for the chain with an initial burn-in period of one million iterations, and interval between samples for 2,000 chains of MCMC.

To detect possible first-generation migrants and their source population we used GENECLASS2 (*Piry et al., 2004*). We used the likelihood-base estimator *L_home* using *Paetkau et al.*'s, *2004* algorithm. *L_home*, is the likelihood of the individual genotype within the population where it was sampled. GENECLASS2 assumes Hardy–Weinberg equilibrium and species, sexual reproduction. The probability of each individual to be encountered in a given population was estimated using 100,000 simulated individuals with a threshold value of 0.01.

## Mating system and contemporary gene flow

We established a $50 \times 20$ m plots ($200$ m$^2$) within four fragments, two medium size fragments and two populations from large fragments. We did not include small fragments due to the lack of enough samples of maternal families (mother and progeny). Plant density was characterized in each population by counting the number of flowering individuals. In each plot, we collected leaf tissue and mature infructescences of maternal plants ($n = 33$). We germinated seeds and collected the leaf tissue from the emergent seedlings for DNA extraction. The number of maternal families varied between populations and each family between 2 and 11 individual plants (Table 2). To estimate outcrossing rate ($t$) and pollen pool structure ($\Phi_{FT}$) we used six highly polymorphic unlinked microsatellite loci (0432, 5409, 1233, 4536, 4483, and 1808) (Table S2) (*Suárez-Montes, Tapia-López & Núñez-Farfán, 2015*) and analyzed all members of each maternal family (6–10 families per population).

We estimated parental inbreeding coefficient ($F$), multilocus outcrossing rate ($t_m$), single-locus outcrossing rate ($t_s$), and biparental inbreeding due to mating among relatives ($t_m - t_s$) using MLTR (*Ritland, 2002*). Standard errors were derived from 1,000 bootstraps. To estimate the confidence interval (CI at 95%) we used the estimated MLTR mean $\pm 1.96 \times 1$ sd. We also estimated the CI for $F$, $t_m$, and $t_s$ by the bootstraping in MLTR (*Ritland, 2002*). Progeny inbreeding was calculated with GENETIX v4.05, based on 10,000 permutations (*Belkhir et al., 1996–2004*).

We estimated the differentiation of pollen pools ($\Phi_{FT}$) sampled by different maternal families using the TwoGener method (*Austerlitz & Smouse, 2002*) as implemented in GenALEx 6.502 (*Peakall & Smouse, 2012*). This model assumes uniform individual distribution and accurate density estimates. If the analysis reveals high $\Phi_{FT}$ then pollen dispersal is restricted, suggesting that different mothers are sampling pollen from, at least partially, non-overlapping sets of fathers. To avoid overestimation of $\Phi_{FT}$ values due to parental inbreeding ($F_p$) and selfing ($s$), we used the formula

$\Phi'_{FT} = \frac{\Phi_{FT}}{1+F_p}$, and given $s$ as $1 - t_m$ where $t_m$ is the multilocus outcrossing rate, $\Phi'_{FT}$ transforms to $\Phi'' = \frac{2\Phi'_{FT} - S^2}{2(1-S)^2}$ for the selfing rate. Finally the pollen dispersal distance ($\delta$), the effective number of pollen donors ($N_{ep} = \frac{1}{2}\Phi''_{FT}$), and the effective pollination neighborhood area, ($A_{ep} = \frac{N_{ep}}{d}$), where $d$ is the density of reproductive plants (*Austerlitz & Smouse, 2001a*; *Austerlitz & Smouse, 2001b*).

**Table 2 Mating system and pollen structure parameters of *Aphelandra aurantiaca* from Los Tuxtlas.**

| Fragment size: | Medium | | Large | |
|---|---|---|---|---|
| Population: | 6 Med | 7 Med | 9 Lrg | 11 Lrg |
| Density/m$^2$ | 1.6 | 0.20 | 0.18 | 0.18 |
| $n$-mothers | 11 | 6 | 10 | 6 |
| $n$-progeny | 91 | 56 | 75 | 32 |
| Parental inbreeding: F (sd) | 0.13 (0.11) | −0.20 (0.01) | 0.10 (0.11) | 0.09 (0.09) (0.03, 0.12) |
| B.C.I | (0.06, 0.19) | (−0.22, −0.18) | (0.04, 0.13) | (0.03, 0.12) |
| Progeny inbreeding: F | 0.13 | 0.03 | 0.03 | 0.04 |
| B.C.I | (0.05, 0.2) | (−0.04, 0.1) | (−0.05, 0.1) | (−0.09, 0.14) |
| Multilocus outcrossing rate: $t_m$ | 0.86 (0.06) | 0.90 (0.05) | 0.67 (0.11) | 1.0 (0.09) |
| B.C.I | (0.80, 0.93) | (0.87, 0.92) | (0.63, 0.73) | (0.89, 1.0) |
| Single-locus outcrossing rate:$t_s$ (sd) | 0.71 (0.07) | 0.85 (0.06) | 0.65 (0.13) | 0.77 (0.11) |
| B.C.I | (0.60, 0.76) | (0.78, 1.0) | (0.61, 0.68) | (0.63, 0.86) |
| Biparental inbreeding: $tm − ts$ (sd) | 0.15 (0.05) | 0.04 (0.05) | 0.02 (0.03) | 0.24 (0.14) |
| P.C.I. | (−0.05, 0.2) | (−0.05, 0.1) | (−0.04, 0.09) | (−0.02, 0.52) |
| Correlation paternity | $\Phi_{FT}$  $\Phi''_{FT}$ <br> 0.30$^*$ 0.49 | $\Phi_{FT}$  $\Phi''_{FT}$ <br> 0.10$^*$ 0.17 | $\Phi_{FT}$  $\Phi''_{FT}$ <br> 0.16$^*$ 0.19 | $\Phi_{FT}$  $\Phi''_{FT}$ <br> 0.26$^*$ 0.40 |
| Effective pollen donors: $Nep$ $\Phi''$ | 1.00 | 2.88 | 2.57 | 1.23 |
| Genetic Neighborhood: $A_{ep}$ (m$^2$) | 0.62 | 14.4 | 14.30 | 6.85 |
| Pollen distance $\delta$ (m) | 0.41 | 1.09 | 1.08 | 1.04 |

**Notes.**

F, inbreeding of progeny (estimated in GENETIX) and inbreeding coefficient of maternal parents (estimated in MLTR); $t_m$, multilocus outcrossing rate; $t_s$, single-locus outcrossing rate; $t_m − t_s$, biparental inbreeding; $sd$, standard deviation 95% bootstrap confidence interval (B.C.I) and parametric confidence interval (P.C.I.); $\Phi''_{FT}$, Correlation of paternity corrected by inbreeding and selfing rate.

$^*p < 0.05$.

## RESULTS

We did not detect evidence of null alleles. Significant deviations from Hardy–Weinberg equilibrium ($p < 0.05$) were observed in all populations, likely due to heterozygotes deficit rather than null alleles. Across populations, 10 loci displayed a significant heterozygotes deficit and one (1810) displayed a significant excess. Exact test of linkage disequilibrium indicated significant deviations at six out of 55 possible primer pair's comparisons. Deviations of Hardy–Weinberg equilibrium could be explained by age structure caused by overlapping generations and patchy plant distribution that can create Wahlund-like effects.

### Genetic diversity and inbreeding

Overall, levels of mean population genetic diversity of *A. aurantiaca* at Los Tuxtlas rainforest were high ($N_a = 5.5$, $H_o = 0.57$, $H_e = 0.61$) and similar among fragments. High genetic diversity values are expected for long-lived perennial plants ($H_o = 0.63$, $H_e = 0.68$; *Nybom, 2004*). Inbreeding coefficient values were low ($F_{IS} = 0.097$, $p = 0.00$), ranging −0.066–0.29 for populations, and statistically similar between fragments of different size (range: 0.05 to 0.114), but slightly lower for large fragments (Table 1). Mantel's test did

not detect a relationship between genetic and geographical distances of all populations ($r = -0.28, p = 0.94$). Geographic distances between populations are given in Table S4.

## Effective population size and bottleneck analysis

Effective population sizes ($N_e$) are relatively large in all fragment sizes (Table 1). Average effective population size of large fragments ($N_e = 205.7$ (CI [123–480.6])) is not significantly higher than in small fragments ($N_e = 119.2$ (CI [86.4–176.7])). However, effective population size tends to decrease with reduction of fragment size. The lowest estimated $N_e$ corresponds to one small fragment (19.2 at fragment 3 SM) while the highest value (infinite $N_e$ estimate) corresponds to a large fragment (9 Lrg).

Under the Stepwise Mutation Model (SMM) and Two-Phase Mutation Model (TPM), the bottleneck test failed to detect a recent genetic bottleneck. The proportion of heterozygotes observed ($H_O$) was lower than expected ($Heq$), suggesting an absence of recent genetic bottlenecks in fragments of all sizes (all $p < 0.001$). All fragments sizes exhibited significant allele deficiency (Table S5).

## Demographic history (ABC)

Population expansion was the best scenario and had the highest posterior probability ($p = 0.99$, 95% CI [0.9997–0.9998]). The PCA representation exhibited a good recovery of the posterior predictive distribution and the observed data (Fig. S1). Under this model, we found evidence of a small population with an effective population size of approximately 1,180 individuals that expanded to a present effective population size of approximately 68,600 individuals. We estimated that this expansion occurred approximately 2690 years before present (Table S3). We found a type I error rate of 0.03, and a type II error rate of 0.02, indicating a statistical strength of 97% and a high degree of confidence for the population expansion scenario.

## Population differentiation

Populations showed a lack of genetic differentiation despite isolation by fragmentation. The hierarchical AMOVA analysis indicated weak genetic differentiation among all fragments ($R_{st} = 0.037, p = 0.00$), with most genetic variance (96.2%) within populations, while only 3.8% of the variance was among populations within fragment size classes.

Bayesian statistical modeling for clustering implemented in STRUCTURE showed the most likely number of clusters at $K = 2$ (LnP $= -11100.43$) and $K = 3$ (LnP $= -10965.43$) (Fig. S2). For $K = 2$, cluster I includes almost all sampled populations while cluster II was composed of only two populations (3 SM and 4 SM). For $K = 3$, cluster I is also composed by almost all populations (1 SM, 2 SM, 5 SM, 6 Med, 7 Med, 9 Lrg, 11 Lrg, and 12 Lrg), while cluster II and III included two main populations (8 Med, 10 Lrg and 3 SM, 4 SM, respectively) (Fig. S2).

As in the STRUCTURE analysis, the constructed network in NetStruct also detected three groups or communities with significant community partitions ($p < 0.05$). The network indicates that these communities are dispersed over the landscape without a clear pattern (Fig. S3A). The assignment of individuals to communities showed that all are composed of individuals from all populations. However, community I was mainly

composed by populations 10 Lrg, 8 Med, 6 Med, and 5 SM; community II includes 7 Med, 1 SM, 3 SM, 4 SM and 11 Lrg populations; and community III is composed by 9 Lrg, 12 Lrg and 2 SM. Moreover, the mean of SAD was low and similar for the three detected communities (I: 0.0012, II: 0.0010 and III: 0.0013), all with a wide left-skewed distribution suggesting moderate strength of association and constant gene flow levels (Fig. S3B). Community II showed slightly lower association without a very wide skewed tail in comparison with the other communities, suggesting gene flow in the past. The coefficient of variation was high for all communities, but the lowest was for community II.

DAPC showed that membership probabilities were higher for individuals in their home population, ranging from 46% to 77% (Table S6 and Fig. S4). Population 3 SM, and 4 SM had the highest membership probabilities for individuals in their home, suggesting higher isolation. DAPC detected 89 admixed individuals: 33 from small fragments, 18 from medium fragments, and 38 for large fragments. The highest number of admixed individuals belongs to population 11 Lrg (15) and population 5 SM (12). The lowest value was for population 3 SM with one admixed individual.

## Spatial genetic structure, connectivity and barriers

The Population Graph consisted of 12 populations connected by 24 edges that exhibited a significant conditional covariance. The topology showed that all populations formed a single interconnected network, indicating gene dispersal (Fig. 2). No IBD (Isolation by distance) among populations was detected (Mantel $Z = 67.8$, $p = 0.805$). Conditional genetic distances (cGD) are shown in Table S4. We found extended edges (between 1 SM–2 SM, 1 SM–3 SM, 1 SM–11 Lrg, 3 SM–4 SM, and 5 SM–6 Med) whose lengths were longer than expected from the spatial distances, indicating long distance dispersal (Fig. 2). We also found compressed edges (between 2 SM–11 Lrg, 3 SM–12 Lrg, 4 SM–7 Med, 4 SM–8 Med, 4 SM–10 Lrg, 7 Med–8 Med, and 11 Lrg–12 Lrg) whose lengths were shorter than expected from the spatial distances, suggesting a reduced permeability of landscape.

## Migration rates and first generation migrants

The BayesAss analysis showed that most gene flow (in the last 2–5 generations) occurred from large to both medium and small fragments (Table 3). However, the large–medium rate was higher than the large-small rate, suggesting higher connectivity between large and medium fragments. GeneClass2 identified 15 putative first-generation migrants out of a total of 388 individuals ($p < 0.01$) (Table S6). These results also indicated considerable gene dispersal from the largest fragment. We found that nine out of the 14 migrants derive from the large fragment, five reside in small fragments, two in medium fragments and two in large fragments. Three derived from medium fragments and reside in smaller populations. Two out of 14 migrants derived from small fragments, one resides in a medium fragment and the other in a large fragment (Table S6).

## Mating system, pollen structure and pollen movement

Plant density of *A. aurantiaca* at Los Tuxtlas is variable among populations. Densities ranged from 0.42 to 3.18 individuals/m$^2$; similar values have been reported by *Calvo-Irabién*

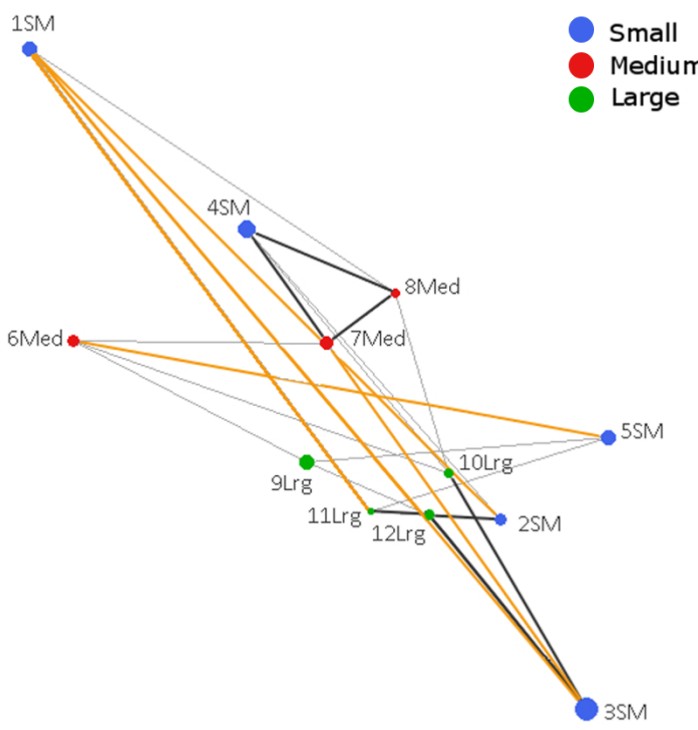

**Figure 2  Genetic network obtained from Population Graph for *Aphelandra aurantiaca* populations at Los Tuxtlas.** The differences in node (circles) size reflect differences in genetic variability within populations. Edge length (lines connecting nodes) represents the among population component of genetic variation. The figure shows normal edges whose length is proportional to that expected under a model of isolation by distance (thin black lines); extended edges (yellow) indicate long distance dispersal events, and compressed edges (thick black lines) indicate reduced gene permeability of the landscape.

**Table 3  Gene flow estimates for *Aphelandra aurantiaca* between small, medium, and large fragments.**

| Pair fragmented populations | | Nm | BayesAss |
|---|---|---|---|
| 1 | 0 | Based on $Fst$ [a] | Short-term gene flow $m$ (95% credible set) |
| Small | Medium | 152 | 0.0459 (0.016–0.075) |
| | Large | 38.5 | 0.0408 (0.011–0.070) |
| Medium | Small | 152 | 0.0042 (−0.0014–0.009) |
| | Large | 37.2 | 0.0149 (0.003–0.026) |
| Large | Small | 35.5 | 0.2069 (0.175–0.238) |
| | Medium | 37.2 | 0.2672 (0.235–0.299) |

**Notes.**
Migration rate ($m$) estimated using BayesAss v 3.0 (*Wilson & Rannala, 2003*) for paired fragment sizes (small, medium, and large). Where $m$ [1][0] is the fraction of individuals in population 0 that are migrants from population 1.
[a] Indirect measures of gene flow ($Nm$) for each paired fragment size were calculated with the formula of *Wright (1951)*: $Nm \approx \frac{1}{4}\left(\frac{1}{Fst} - 1\right)$.

*(1989)*, who found a range from 0.38 to 3.32 individuals/m². The density of reproductive individuals in our study ranged from 0.18 to 1.6 individuals/m².

*Aphelandra aurantiaca* has a mixed mating system ($t_m = 0.81$), although mating system estimators of each population are not related to fragment size (Table 2). There were no significant differences among populations in multilocus outcrossing rate, single outcrossing rate, and biparental inbreeding. Bootstrapped confidence intervals showed a significantly lower multilocus outcrossing rate in one population (11 Lrg). The AMOVA of gametes indicated that most genetic variance was contained within mothers (70–90%). The correlation of paternity estimates($\Phi_{FT}$), were high and significantly higher than zero for all populations (Table 2). Hence, the correlation of paternity estimates,$\Phi_{FT}$, indicated restricted pollen dispersal (part of the offspring are full-sibs), with a relatively low number of effective pollen donors ($N_{ep}$), and short pollen distance movement in this species (Table 2). Some populations showed a high (45%) or a low (17–19%) fraction of siblings sharing the same father. The number of effective pollen donors ($N_{ep}$) was relatively low (1.92, on average) (Table 2). Average inbreeding values [$F_{is}(s.d.)$] were slightly lower for adult maternal plants [0.03(0.15)] than for seedlings [0.05(0.04)]. Furthermore, we detected biparental (or uniparental) inbreeding in all populations ($t_m - t_s$ ranged from 0.02 to 0.24), that did not differ between populations. The effective pollination neighborhood ranged from less than 0.6 to 14.4 m², whereas pollen distance movement ranged from half a meter up to 1.1 m (Table 2). We suggest caution in the interpretation of TwoGener results due to uniform individual distribution assumptions.

## DISCUSSION

### Genetic structure and habitat fragmentation

Habitat fragmentation has not produced genetic differentiation or immediate reductions in genetic diversity of *A. aurantiaca* despite dozens of generations since the onset of fragmentation in the region of Los Tuxtlas. Regardless of fragment size, populations possess private alleles and high genetic diversity ($H_e = 0.61$), similar to those of long-lived perennial herbs (*Nybom, 2004*), but higher compared to other understory Acanthaceae plants [e.gr., *Graptophyllum reticularum* ($H_e = 0.31$), *Graptophyllum ilicifolium* ($H_e = 0.43$) and *Ruellia nudiflora* ($I = 0.26$) (*Shapcott, 2007*; *Vargas-Mendoza et al., 2015*). Effective population size estimates ($N_e$) showed that most populations are effectively large, suggesting that habitat fragmentation has not as yet reduced $N_e$ enough to detect an impact on genetic diversity.

The weak genetic structure detected ($R_{st} = 0.037$) is supported by a number of analyses, suggesting that *A. aurantiaca* populations have remained genetically connected. Most genetic variation is contained among individuals within populations (96%) rather than between populations (3%) in fragments of different size. A similar result was obtained for other understory tropical herbs in the same region (*Cuartas-Hernández & Núñez-Farfán, 2006*; *Suárez-Montes, Fornoni & Núñez-Farfán, 2011*). Genetic clustering analyses revealed that most populations shared genetic information of one cluster or community, which explain the low genetic differentiation. However, small populations (3 SM and 4 SM) form a genetic group, suggesting ongoing isolation.
The high genetic diversity, large effective population size, and low genetic differentiation found in populations of *A. aurantiaca* could be related to historical processes of populations rather than with the present landscape configuration. Also, life history characteristics of *A. aurantiaca*, such as its mating system, generation time, vegetative reproduction, and overlapping generations, might help to diminish the impact of genetic drift, maintaining large effective population size, and buffering the loss of genetic diversity due to habitat fragmentation (*Weidema, Magnussen & Philipp, 2000*; *Hailer, Helander & Folkestad, 2006*; *Breed, Christmas & Lowe, 2014*; *Pellegrino, Bellusci & Palermo, 2015*).

## Genetic structure, gene flow and past demographic change

Although low genetic differentiation and occasional long-dispersal events of *A. aurantiaca* were detected, we also found evidence of restricted gene flow. The contrasting results of restricted ecological dispersal of *A. aurantiaca* over short distances and low genetic structure could indicate a lack of population equilibrium under current demographic conditions. When populations suffer frequent extinction and re-colonization processes, low *Fst* values are expected if colonist individuals are drawn from distant populations. Besides, since dispersal could be highly variable through time, direct measures of dispersal could miss long distance dispersal events (*Coyne et al., 1982*; *Slatkin, 1985*; *Slatkin, 1994*; *Whitlock & McCauley, 1990*).

Historical data suggest ancient contraction-expansion of Los Tuxtlas rainforest. Contractions occurred during a period of low temperatures and humidity (from 20,000 to 12,000 years ago during the last glacial maximum (LGM)) followed by subsequent vegetation expansion events (*Graham, 1975*; *Toledo, 1982*; *Haffer & Prance, 2001*; *Gutiérrez-Rodríguez, Ornelas & Rodríguez-Gómez, 2011*). In *A. aurantiaca*, ABC analyses suggest a plausible past population expansion scenario at Los Tuxtlas around the end of the LGM, when warmer climatic conditions established. More recently, contraction-recolonization of Los Tuxtlas rainforest could also be related with volcanic activity (during the last 153 years ago) (*Martin Del Pozzo, 1997*; *Guevara & Laborde, 2012*), and with forest fragmentation due to human activities (only during the last 42 years) (*Dirzo & Garcia, 1992*). Further evidence indicates no relationship of geographic and genetic distances among populations of *A. aurantiaca*, suggesting a relatively recent origin. There is also evidence of ancient and recent population expansion for an abundant palm of the understory of Los Tuxtlas (J Juárez–Ramírez, 2015, unpublished data; *Martínez-Ramos et al., 2016*). Therefore, the low genetic structure of *A. aurantiaca* could be due to different historical processes at Los Tuxtlas rather than recent habitat fragmentation.

## Current gene flow and habitat fragmentation

The highest rates of migrants and first-generation migrants derived from the largest and medium fragments, underscore the importance of relatively large forest patches to prevent genetic isolation. Moreover, pollinators may use a series of different fragment sizes to forage, helping to maintain connectivity across the landscape (*Llorens et al., 2012*; *Volpe et al., 2014*). Specifically, hummingbirds are effective pollinators that can fly across relatively large areas during their foraging routes, carrying pollen grains to individuals' located far

apart (*Stouffer & Bierregaard, 1995*; *Kraemer, 2001*). However, it is necessary to conduct specific studies to assess the effect of Los Tuxtlas forest fragmentation on the abundance and behavior of the hummingbird pollinator *Phaethornis longirostris,* and their consequences on reproductive output of *A. aurantiaca.*

Although both pollen and seed dispersal are relevant to the pattern of genetic structure in *A. aurantiaca,* analyses do not allow us to determine which process is the most important contributor to gene flow. Natural gene flow often follows a leptokurtic distribution, implying that most genes move over short distances and only a small fraction move over long distances. Pollen dispersal kernels are often short, resulting in self-pollination or gene exchange among closely related individuals (*Betts et al., 2014*; *Ellstrand, 2014*). However, even a small number of long distance migration events can suffice to reduce $F_{st}$. Unfortunately, these rare events are difficult to detect in field studies (*Nathan et al., 2003*; *Mona et al., 2014*). For *A. aurantiaca*, it is likely that current events of long distance dispersal contribute to maintain the landscape connectivity preventing genetic differentiation and increasing local genetic diversity. In contrast, restricted pollen (0.41–1.09 m) and seed dispersal (1.5 m; *Calvo-Irabién, 1989*) promote substructure within populations.

Compressed edges suggest reduced gene permeability among populations that are geographically close, even in the large fragment (Fig. 1). Within the large fragment only population 12 Lrg shows contact with three other populations, but one of them is a compressed edge (11 Lrg–12 Lrg). This could be related to the physical barrier imposed by the "Vigia" hill (ca 600 m a.s.l.) within the preserve, reducing gene dispersal. Thus, topography (elevation) of the landscape should be considered in future studies as a factor affecting gene flow.

## Mating system and fragmentation effects

The species' mating system is an important factor that affects the distribution of genetic diversity. *Aphelandra aurantiaca* is predominantly outcrosser ($t_m = 0.81$). The description of *A. aurantiaca* as a selfing species with a mixed mating system agrees with values found in other hummingbird-pollinated plants in the Neotropics (*Wolowski et al., 2013*). However, we found that its mating system is predominatly of outcrossing. A mixed-mating system can combine advantages of both reproductive strategies: outcrossing promotes genetic diversity when pollinators are abundant, while self-fertilization may ensure reproduction when pollinators are scarce or absent (*Goodwillie, Kalisz & Eckert, 2005*; *Ruan & Teixeira da Silva, 2012*).

Self compatible tropical herbs do not necessarily suffer of inbreeding because they may possess breeding system traits that promote outcrossing (*McDade, 1985*). In general, *A. aurantiaca* showed no signs of inbreeding, although some populations exhibited inbreeding (Table 1). This finding could be a consequence of the Wahlund effect caused by genetic structure within populations (*Murren, 2003*) due to limited seed/pollen dispersal within populations. For *A. aurantiaca* at Los Tuxtlas fragmented rainforest, maintaining a mixed mating system with a high outcrossing rate appears to help preventing the loss of genetic variation, as would be theoretically expected for smaller populations.

Forest fragmentation does not seem to affect the contemporary pollen dynamics of *A. aurantiaca*, as in other herbs at Los Tuxtlas (*Cuartas-Hernández, Núñez-Farfán & Smouse, 2010*). The pollen pool structure was variable; in some populations it was more restricted than in others. The effective pollination neighborhood estimated resulted smaller than 14 m$^2$, and plants received pollen from neighbors located, on average, within a radius of 1.1 m. This finding agrees with an estimation of pollen movement using fluorescent dyes (reported by *Calvo-Irabién, 1989*). The effective number of pollen donors of *A. aurantiaca* is relatively low (range 1.0–2.8), suggesting that the potential pollen donors contribute little to $N_{ep}$. The moderate biparental inbreeding in *A. aurantiaca* could be explained by the limited seed dispersal and mating among close relatives, which may be due to hummingbirds' moving among relatively close plants (P Suárez-Montes, pers. obs., 2014). High plant density may contribute also to increase inbreeding and shorten pollen dispersal distance in *A. aurantiaca*. However, assessing whether high plant density reduces pollen dispersal makes necessary an extensive sampling of populations with different densities. We suggest caution when interpreting results of TwoGener given that it assumes uniform individual distribution.

## CONCLUSIONS

Current genetic structure of *A. aurantiaca* is the result of different factors acting simultaneously. Despite extensive forest fragmentation of Los Tuxtlas rainforest, *A. aurantiaca* maintains high genetic diversity and low genetic differentiation between populations, suggesting effective gene flow. Habitat fragmentation has not affected the outcrossing rate and pollen dynamics within populations. Demographic history and life history characteristics are important to explain the current pattern of low population structure of *A. aurantiaca*, rather than recent fragmentation effects. We propose that past demographic dynamics, large effective populations, long distance gene dispersal events, and life history characteristics of this species, such as mixed mating system, overlapping generations, and ability to re-sprout after forest disturbance (e.g., light-gaps formation), ameliorate the effects of fragmentation. In addition, higher gene flow originated from medium and large fragments favour genetic connectivity and confirm their importance as genetic reservoirs and gene sources. Conservation efforts must be directed to preserve these fragments. However, small fragments should not be overlooked, as they may act as stepping stones to increase/maintain connectivity among fragmented populations, especially for species whose gene flow is aided by animals. Our findings should contribute significantly to the development of effective conservation strategies for *A. aurantiaca* and species with similar mating systems and pollinators.

## ACKNOWLEDGEMENTS

The authors thank to the staff of the Los Tuxtlas Biological Research Station and Rosamond Coates who allowed us the use of the facilities. We also thank Rosalinda Tapia López for lab advice, to Guadalupe Andraca Gómez, César A. Domínguez, Victor Parra Tabla, and Edson Sandoval for valuable suggestions to an early draft of this paper, and Helen Salazar

Arteaga and Javier Laborde Dovalí for assistance in landscape description. This study is part of the PhD Dissertation of P Suárez-Montes in the Graduate Program in Biological Sciences, UNAM.

### Funding

The project "Evolutionary ecology studies of rainforest fragmentation in Mexico" was funded by a Papiit grant, UNAM (IN 215111-3), to JNF. The funders had no role in study design, data collection and analysis, decision to publish, or preparation of the manuscript.

### Grant Disclosures

The following grant information was disclosed by the authors:
Papiit grant, UNAM: IN 215111-3.

### Competing Interests

The authors declare there are no competing interests.

### Author Contributions

- Pilar Suárez-Montes conceived and designed the experiments, performed the experiments, analyzed the data, contributed reagents/materials/analysis tools, wrote the paper, prepared figures and/or tables, reviewed drafts of the paper.
- Mariana Chávez-Pesqueira performed the experiments, contributed reagents/materials/analysis tools, reviewed drafts of the paper.
- Juan Núñez-Farfán conceived and designed the experiments, performed the experiments, contributed reagents/materials/analysis tools, wrote the paper, reviewed drafts of the paper.

### Data Availability

The raw data has been supplied as a Supplemental File.

### Supplemental Information

Supplemental information for this article can be found online at http://dx.doi.org/10.7717/peerj.2764#supplemental-information.

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
