# Peer review of "Life history and past demography maintain genetic structure, outcrossing rate, contemporary pollen gene flow of an understory herb in a highly fragmented rainforest"

_PeerJ, doi:10.7717/peerj.2764_

## Round 0.1 · original submission · Major Revisions

I would not require, as per Reviewer #2's suggestion, that the ABC analysis be dropped. However, the manuscript should be clear about the timescale relevant to each analysis, distinguish conclusions about historical structure (including the purported population expansion, which could explain the lack of historical structure) from conclusions about the effects of recent fragmentation, and possibly remove analyses that are not directly relevant to the stated goals of the study.

Also, I was not clear on the rationale for lumping estimates of gene flow between sets of populations belonging to fragments of different sizes (as reported in Table 2).

Throughout the manuscript, please cite the baseline for comparison when saying a statistic is low or high relative to some external comparison.

A list of specific comments by line number is below. Typos and language suggestions are denoted by "existing text" -> "suggested text".

70 "its" -> "theirs"
104 strike "represent"
108 "reveled" -> "revealed"
128 what is "structuring of the pollen pool"?
195 "parameters settings" -> "parameter settings"
197 "assuming generations of 1 year, and generation time of 1 year" redundancy
208 "program GenALEx 6.502 program" redundancy
213 "based model" -> "model-based"
220 explain the purpose of NetStruct algorithm. what are the nodes and edges?
326 "lower" -> "lowest"
331 Define SSM (stepwise) and TPM (two-phase) mutation models.
343 overly precise. Use confidence intervals?
353 "fragments" -> "fragment"
355 "that the most likely" -> "the most likely"
370 "resulted" -> "was"
376 "higher" -> "highest"
382 spell out "IBD" on 1st usage
395 "resulted" -> "was"
405 "at the Los Tuxtlas" -> "at Los Tuxtlas" - it is redundant to use "the" before "los"
409 "tm" should be subscripted (as with all occurrences of "tm" and "ts" in this section)
410 "showed" -> "show"
412 "bootstraps" -> "bootstrapped"
413 "showed significantly" -> "showed a significantly"
414 "it" -> "they"
422 "lower" -> "low"
423 "low" -> "lower"
424 "biparental inbreeding (or uniparental)" -> "biparental (or uniparental) inbreeding"
430 subscripts missing in the math notation throughout this pgph
431 "A. aurantica have passed" -> "A. aurantica that have passed"
432 "at the region" -> "in the region"
432 "population possesses" -> "populations possess"
443 "for others understory" -> "for other understory"
494 remove commas from both sides of "large and medium size fragments"
495 I cannot parse the sentence starting "These results..."
498 "landscape's physical barriers detected" -> "physical landscape barriers"
514 "numbers" -> "number"
567-571 I'm not sure these conclusions are supported by the results
575 "for allowed use the facilities" -> "who allowed us use of the facilities"
577 "to early draft for" -> "to an early draft of", "The family" -> "Thanks to the family"
810 drop "the"
827 "nodes" -> "node"
829 strike "due to the connecting nodes"
830 "isolation distance" -> "isolation by distance"
831 Table 1: spell out and define how SD is reported. Is it 1 standard dev?
855 "estimated of" -> "estimates in"
In Table 2: include a space before parentheses. Why report SD when you are also reporting the (more useful) 95% credibility interval?
530 "an theoretical expectation" -> "as would be theoretically expected"
553 strike "of", "at" -> "in", and "The"->"the"
555 "Habitat frag. has not affected the outcrossing rate and pollen dynamics". Should causality be considered in both directions?
561 I would have expected the results regarding population expansion (i.e. nonequilibrium pop structure) to inform this conclusion.
565 it is not clear to me how medium and large fragments "favour long dispersal events"
In Table 3, use a space before parentheses. Report units for genetic neighborhood. Clarify the difference between the two F estimates in the footnotes.

Reviewer 1 ·

Basic reporting

I consider this manuscript has been carefully prepared. The English language is mostly correct (but see below my minor comments) and the background provided synthesizes adequately the literature on the genetic consequences of habitat fragmentation in plants. The figures and tables, both the ones included in the main document, as well as the supplementary materials, are relevant and are clear and well prepared. The raw data have been adequately provided.

Minor comments:
L35 assessed
L78 increasedincrease
L79 delete “experience”
L103 constitute represent –delete one of the two words
L197 generations of 1 year, and generation time of 1 year
L213 as a model based approach
L273 under Paetkau's et al. (2004) algorithm
L316 ranging from -0.066 to 0.29
L400-403 delete parentheses
L410 did not show
L431 tens dozens
L432 populations posses
L433 delete “values”
L443 for other
L530 a theoretical

Experimental design

The knowledge gap, the research question and the contribution of the study are clearly identified. Indeed, in my opinion, this study makes a significant contribution by analyzing the effects of habitat fragmentation in a tropical understory herbaceous plant, which is a poorly studied life form in this kind of studies. All stages of the study –design, field work, laboratory procedures and data analyses seem to have been rigorously performed. Data analyses are extensive and through. However, some details require attention:
1. L158-L162 The number and identity of microsatellite markers assayed should be clearly stated.
2. L303-L305. Similar comment. The paragraph is confusing. At the end how many loci were used? Also, the sentence “One SSR locus (1071) had a high probability of being null allele (0.23) and potentially under selection” is problematic. First, microsatellite loci contain but not ARE null alleles. Second, what is the number in parenthesis a probability value or the frequency of the presence of null alleles. Please clarify and explain how that value was obtained. Third, I doubt the validity of the selection test if the locus contains null alleles in high frequency.
3. L328-330. The authors find a positive correlation between effective population size and allelic diversity. However, since effective size is indirectly calculated from genetic variation, it is probable that the correlation is circular.

Validity of the findings

The conclusions are robust and clearly based on the results. My only comment refers to L532-536 where the authors seem to be confusing “inbreeding” with “inbreeding depression”.

Reviewer 2 ·

Basic reporting

The manuscript could be shortened considerably and needs copy editing to tighten up the text and fix several grammatical errors. Several sections are confusing and ambiguous, and thus need to be rewritten or eliminated.

Figure and table legends need to include more detailed information (see general comments).

Experimental design

The relevance of conducting this study on herbs is not clearly stated. On lines 108-112 the authors present an interesting dichotomy where fragmentation may impact gene flow differently in herbs in contrast to other species, however this topic is not explored further in the introduction nor in the rest of manuscript. Aside for contributing to the scant literature on herb gene flow, its contribution to the fragmentation or gene flow literature is limited. The authors should explore if the effects of fragmentation on herbaceous species are comparable to those observed on other plant systems.

Landscape appears to be an important factor shaping gene flow patterns in herbaceous species, however this issues is ignored in the introduction as well as in the methods section and discussion.

The objectives and predictions compound the effects of fragmentation expected by a reduction on effective population size with those expected by changes in pollinator behaviour or densities. Lower Ne is expected to reduce diversity through drift, however changes in mating system parameters and gene flow patterns are more likely attributed to changes in pollinator abundance or behaviour. These concepts are not explored in the manuscript. The discussion on pollinators and seed dispersal patterns is limited or completely absent.

It is important to stress that the manuscript is geared towards investigating contemporary gene flow patterns, while some of their methods aim to analyze historical patterns of genetic structure.

Sites are poorly described. The authors fail to mention the criteria used to select the different forest fragments, nor describe them in detail. How is the forest structure within these sites? Are they all comparable or do they differ in regeneration time and canopy gap openness (important criterium for Aphelandra). It is also unclear why some fragments were not included in the analysis or how why Lrg populations weren’t all grouped into a single site. A most important flaw is that the landscape surrounding the fragments is not described at all. The landscape could significantly impact gene flow dynamics and thus should be at least described or more likely included in the analysis.
The procedure describing the collectection of individual plants and families within populations was also not described.

ABC analyses operate on a different time scale than the other tests performed by the authors. The effects and scenarios observed by ABC analyses probably predate the fragmentation events that shape contemporary gene flow patterns. I suggest removing this analysis as it only contributes to the length of the manuscript.

The authors use several statistical techniques to investigate genetic structure on these populations, however they fail to clarify why using such a large array of analysis, especially if some of them present redundant or contradictory results. Additionally, there are details missing in the description of some methods, for example the AMOVA analyses (see general comments). A similar concern may be raised for gene flow estimates. For example, authors fail to discuss differences between BayesAss and GeneClass results which rely on different assumptions and produce different estimates.

Validity of the findings

Their results are somewhat contradictory; their data suggests localized gene flow with occasional long distance gene flow events, but found no evidence of structure among populations. The authors argue that occasional long distance gene flow events are responsible for reducing structure among patches. Another plausible explanation would suggest that the population is very large, and fragmentation has not yet reduced Ne enough to actually have an impact on genetic diversity. This is consistent with Ne estimates and would hold up even if gene flow distances are limited. However this is not identified as a possible scenario.

Additional comments

Numbers refer to line numbers on the original manuscript.

70. “Reduced to half of their original area”
103. Remove “constitute”
110. Not clear what you mean by: “while in others it varies with landscape context”. This sentence should be clarified. This sentence could also present a testable hypothesis for this paper.
141. How did you avoid collecting genets? Unidentified clones could underestimate pollen flow distances.
146-156. You need to describe the landscape a bit further, perhaps include a map with land use information. You also need to explain the criteria used for site selection. Why were some forest patches excluded from the analysis? Please describe the forest structure (e.g., gap density, forst height, species composition) of the patches. It appears that Large populations are all part of a single forest patch. why did you consider them different populations?
154. How were adults selected within patches?
187. ABC analysis should be removed as they analyze demographic processes over longer periods of time than the ones analyzed with other techniques. Fragmentation, as you state on line 149, occurred recently and its effect on genetic diversity is unlikely to be detected properly by ABC analyses.
197. How did you arrive at generation time of 1 year if this species may live up to 8 yrs?
201. It is unclear how you use PCA to test for model fit, as this is an ordination technique.
207. Arlequin is better suited for AMOVA and it is considered the gold standard. You should probably repeat the analysis with that program. It is unclear why you include the “individual level” if you don’t have gametic phase info. For the AMOVA you also need to describe the hierarchical structure and significance tests.
218. Structure harvester has problems identifying the correct K when K=1 or K=2. Pritchard’s likelihood procedure should also be used. Did you consider the possibility that you have only one cluster (K=1)?
220. This sentence is confusing, please consider rewriting it.
223. Please define SAD and SA in detail and how the procedure to test for statistical significance. This whole section is unclear and should probably be rewritten.
272. What is the L_home statistic?
273. “Using Paetkau et al.’s algorithm”
274. Explain the relationship between BayesAss and GENECLASS2 estimates and why are they both needed.
277. It is unclear why you use 6 out of 10 primers. Why the subset? Please clarify how many loci were used on all analyses. It is also important to show the diversity of this six loci, since that directly influences MLTR estimates.
281. How did you collect the families? Please clarify. Did you use seeds or seedlings? Why didn’t you include families from the smaller fragments?
299. Nep estimates based on \phi_FT rely on several unrealistic assumptions such as homogeneous distribution, accurate density estimates, etc. Please address this issues here and in the discussion section.
310. Did you discard loci in LD? How did you deal with LD?
311. Not sure why you have 47 pairwise comparisons. If you have 10 loci you have (10x9)/2 primer pairs: 45. Please explain these results.
316. You did not test if the inbreeding coefficient was significantly different from 0.
323. Consider rewriting this sentence.
325. How did you assess differences in Ne? Based on CI overlap, Ne appears to be similar across size patches.
325-335. These results suggest that fragmentation has not yet reduced N_e in the smaller patches. Populations appear to belong to a larger panmictic population, genetic diversity and structure may have not yet suffered from anthropogenic effects of habitat fragmentation.
337. Population expansion 3000 years ago is hardly related to the effects of habitat fragmentation in the last 50 yrs. THis analysis contributes very little to your manuscript.
350. Significance of Rst value?
361. I do not see any clear structure pattern. Did you consider the possibility of a single cluster?
368. It is unclear how you interpret SAD values.
373. Based on figure S4, DAPC analysis correlate with STRUCTURE results; they do not indicate a clear pattern of population structure. Eightynine admixed individuals out of 388 is about 22% of admixture which corresponds well with BayesAss estimates, this suggests high rates of gene flow and could eliminate structure.
380. What does “significant conditional covariance” mean?
387. Compressed edges between SM and Lrg fragments suggest that gene flow between large and small fragments is uncommon (i.e., reduced permeability). Doesn’t this contradict some of your other results? Graph analysis correlate with lack of structure and ample gene flow, as in your previous analyses.
400. Your conclusion about skewed gene flow is based on 15/388 (3%) migrants. You should reconsider this result as a key component of your discussion.
426. How do you explain your structure results given this limited pollen movement. Seed dispersal should also be very limited (ballistic-gravity dispersal). You should address this apparent discrepancy in the discussion, mention the role of biotic gene vectors on gene flow and genetic structure estimates.
445-449. This sentence is poorly written and it difficult to understand what the authors want to convey. I do not see any genetic groups that could be interpreted given the current patch configuration.
450-452. This is the most likely explanation for your results; a large population that has not yet revealed the negative effects of fragmentation.
459-466. This paragraph is unclear. Please consider revising it.
471. A “propagule model” where all patches contribute to a newly colonized population would produce similar results. Do you have any evidence supporting a non-equilibrium metapopulation? Do you have any information regarding pollen or seed vectors or gene movement patterns for this species? Fragmentation could have affected pollinator behaviour or densities reducing gene flow distances. Your estimates are presumably based on a single generation (you never mentioned if you collected seeds) and this could reflect a contemporary effect of fragmentation which could affect structure in the following generations. Conversely, if gene flow distances vary widely among years, you could just have data on a year when gene flow was more localized than on any average year. Do you have any information that could shed some light on any of these scenarios?
478-483. I don’t see how you can relate population expansion in the last 3K yrs with fragmentation in the last few decades. Time frames are way off!
489. This section should be reduced considerably given the limited number of migration events observed.
498. What “landscape’s physical barriers” are you referring to? Do you have extensive or limited gene flow? It is unclear what you want to achieve with this paragraph.
506. If elevation seems to be a factor, why did you not included in the analysis? You need to go back and describe the landscape in more detail and probably analyze your data taking landscape features into account.
513. In this case I would argue that long distance gene dispersal is more likely to be attributed to pollen flow. Hummingbirds are more likely to disperse pollen over larger distances in contrast to ballistic/gravity seed dispersal. There are many papers describing gene flow patterns for hummingbird pollinated species which could assist you in your discussion.
516. Long distance events should have an effect on average gene flow distances. Sub-structuring should have been picked up by STRUCTURE. Can you perform a SGS analysis?
536. What is it, low or high inbreeding? YOur data suggest NO inbreeding.
537. This substructure should have potentially been picked up by STRUCTURE.

Figure 1. This map should include land use information or more information on the landscape.

Table 2. “Gene flow estimates for Aphelandra aurantiaca between small, medium and large fragments”.
Table 2. Legend should have more info: stats, program, site, etc.

---

## Round 0.2 · accepted · Accept

I appreciate the detailed response to the reviewers' feedback, and judge that their concerns have all been adequately addressed.

Reviewer 2 ·

Basic reporting

The manuscript has improved considerably from the last version. However, it still requires some copyediting by a native english speaker, to improve clarity.

Experimental design

The original problems with the experimental design have been improved.

Validity of the findings

Results are clearly stated. I still think that the relationship between the findings from ABC analyses and those from contemporary estimates of gene flow, should have been stated more clearly. Population expansion is compatible with the lack of observed genetic structure.